# ICD-10 Codes to Identify Adverse Drug Events Associated with Antibiotics in Administrative Data

**DOI:** 10.3390/antibiotics14030314

**Published:** 2025-03-18

**Authors:** Hannah Lishman, Amber Cragg, Erica Chuang, Carl Zou, Fawziah Marra, Jennifer Grant, David M. Patrick, Corinne M. Hohl

**Affiliations:** 1BC Centre for Disease Control, Vancouver, BC V5Z 4R4, Canada; hannah.lishman@bccdc.ca (H.L.); erica.chuang@bccdc.ca (E.C.); carl.zou@bccdc.ca (C.Z.); jennifer.grant@bccdc.ca (J.G.); 2Department of Emergency Medicine, Faculty of Medicine, University of British Columbia, Vancouver, BC V5Z 1M9, Canada; amber.cragg@ubc.ca (A.C.); corinne.hohl@ubc.ca (C.M.H.); 3Faculty of Pharmaceutical Sciences, University of British Columbia, Vancouver, BC V6T 1Z3, Canada; fawziah.lalji@ubc.ca; 4Department of Pathology and Laboratory Medicine, Faculty of Medicine, University of British Columbia, Vancouver, BC V6T 1Z7, Canada; 5School of Population and Public Health, Faculty of Medicine, University of British Columbia, Vancouver, BC V6T 1Z3, Canada; 6Vancouver Coastal Health Research Institute, Vancouver, BC V5Z 1M9, Canada

**Keywords:** antibiotics, adverse drug events, hospitalization, emergency department, kappa

## Abstract

Antibiotics are among the most used therapeutics in primary care, and while their benefits are clear, the potential harms related to adverse drug events (ADEs) cannot be ignored. We outline the creation of a comprehensive list of diagnostic codes describing antibiotic-associated ADEs resulting in presentations to acute care hospitals. **Methods:** Previously published ADE codes were used to link BC hospitalizations to prior outpatient antibiotic prescriptions and were restricted based on whether patients received an antibiotic within a month prior to the ADE-related hospitalization. The code list was reviewed by two clinical experts independently for the likelihood of being antibiotic-associated. The inter-rater reliability was calculated using Kappa scores with 95% confidence intervals (CIs). **Results**: Of the 695 ICD-10 ADE codes with evidence of recent antibiotic administration, 72, 68, and 555 codes were considered likely, possibly, and unlikely antibiotic-associated, respectively. **Conclusions:** We outline a methodology for developing an ICD-10 code list for antibiotic-associated ADEs severe enough to warrant hospital admission. This will help to improve the use of administrative data to capture antibiotic-associated ADEs.

## 1. Introduction

Outpatient antibiotic use is common, but may confer health risks that compromise its therapeutic benefits, including allergic reactions (i.e., anaphylaxis and skin rash), disruption of the gut microbiome (i.e., Clostridioides difficile infection) and drug interactions [1,2,3]. Adverse drug events (ADEs) associated with outpatient antimicrobial medications are responsible for more than 280,000 patient visits to emergency departments and 26,000 admissions across Canada per year, resulting in over USD $400 M in annual healthcare expenditures [4,5,6,7,8]. These costs do not take into account the additional burden of antibiotic-resistant infections and attributable deaths. The Council of Canadian Academies reported that in 2018, 26% of the approximately 1 million bacterial infections in Canada were resistant to first-line treatments, with resistant infections being responsible for over 14,000 deaths, 5400 of which were directly attributable to antimicrobial resistance [9]. Capturing data on antibiotic-related adverse drug events observed in clinical practice is important to understand the real-world clinical benefits and harms of antibiotics, and inform drug regulation, prescribing guidelines, and efforts to curb clinical harm and antibiotic resistance.

Administrative databases are readily available, inexpensive and can provide population-level health data on important health outcomes. In Canada, the Discharge Abstracts Database (DAD) captures hospital administrative data including up to 25 hospital diagnoses [10]. In the past, we produced a list of ICD-10 codes to identify ADE-related emergency department visits [11] and evaluated the code set for its completeness in identifying a broad range of ADEs [12]. In this work, we aim to identify a list of ICD-10 codes to identify ADEs specifically related to outpatient antibiotic dispensations that are severe enough to require hospitalization.

## 2. Materials and Methods

### 2.1. Identifying Potential Antibiotic-Associated Adverse Drug Events

We focused on adverse drug events (harm caused by appropriate or inappropriate use of a drug) as opposed to adverse drug reactions (harm caused by a drug under appropriate use) as empiric use of antibiotics is common practice and therefore assessing appropriateness is often not possible based on a lack of laboratory data. We identified potential antibiotic-associated ADEs by linking British Columbian administrative hospitalization (DAD) data with provincial medication dispensing data (PharmaNet) from 1 January 2001 to 31 December 2020 [13,14]. We started by identifying all hospital healthcare encounters coded with an ADE-related ICD-10 diagnostic code with the same first three digits as those identified by Hohl et al. in a systematic review of the literature [11,12,15].

We excluded ADE-related diagnostic codes describing events that occurred during a patient’s hospital stay (i.e., diagnosis type 2 codes—“post-admit comorbidity—conditions that arise following admission and satisfy the requirements for determining comorbidity”) [16]. We then excluded all healthcare encounters that were not preceded with one or more outpatient antibiotic dispensations within 30 days of the healthcare encounter. Analyses were performed using R Studio 2023.12.0.

This restricted code set was then assessed by two independent reviewers for the likelihood of the ADE being antibiotic-associated. A physician with expertise in infectious diseases and microbiology (JG) and a clinical pharmacist (FM), both with extensive clinical experience practicing in British Columbia, independently rated the certainty by which any antibiotic could be responsible for the ICD-10 diagnosis coded in a subsequent hospital visit. Reviewers considered ADEs that could be directly (i.e., toxicity or drug interaction) or indirectly attributed to the use of antibiotics (i.e., dysbiosis or drug interaction). Reviewers used the Hohl et al. coding system (Table 1) [11]. Any discrepancy in the assigned causality rating for each ICD-10 code was discussed between the two reviewers, and a third clinical reviewer (CMH) where necessary, until consensus was reached.

### 2.2. Rating Inter-Rater Reliability

We calculated a raw and a weighted Cohen’s Kappa statistic to determine inter-rater reliability of the two reviewers’ initial ratings prior to reaching consensus. Causality codes were categorized into Likely (A, B, or C), Possible (D or E), and Unlikely (U or V). “Categorical agreement” was defined as agreement across the Likely, Possible, and Unlikely categories and “essential agreement” was defined as agreement across the A1, A2, B1, B2, C, D, E, U, and V categories. To calculate the weighted Kappa statistic, a linear weighting matrix was used to weigh differences between Likely and Unlikely more heavily than differences between Likely and Possible. Both raw and weighted Kappa statistics were calculated based on categorical agreement.

## 3. Results

Among the original list of 827 ICD-10 codes used in the literature for identifying ADEs, 695 different ADE-related codes were identified by linking provincial administrative hospital admission data with 30-day antibiotic dispensation data. There were 99 essential discrepancies (14.2%) and 97 categorical discrepancies (14.0%) between the two raters prior to reaching consensus. The unweighted categorical agreement between the two reviewers was “weak” (k = 0.54, 95% CI 0.45–0.62), but the weighted categorical agreement was “moderate” (k = 0.60, 95% CI 0.52–0.68) [17]. After facilitated discussion and an additional round of review, complete consensus on the code list was reached.

We identified 72 ICD-10 codes (10.4%) that “Likely” identified antibiotic-associated ADEs (categories A1, A2, and B1 and B2 and C) (Table 1). Another 68 ICD-10 codes (9.8%) that “Possibly” identified potential antibiotic-associated ADEs (categories E and D). The remaining 555 ICD-10 codes (79.9%) were deemed “Unlikely” to characterize an antibiotic-associated ADE (categories U and V). Some illustrative examples of “Unlikely” diagnostic categories included, but were not limited to, gastrointestinal ulcers (K25–K28), mental or behavioral disorders due to specified drugs (F11–F19), or external cause of morbidity codes specifying non-antibiotic drugs (Y41–Y88).

The “Likely” and “Possible” ICD-10 codes themselves can be found in Table 2 and Table 3, respectively.

## 4. Discussion

Our objective was to identify an ICD-10 code set among patients admitted to hospital for an ADE related to antibiotic use within the prior 30 days. Two expert reviewers adapted a previously published causal ranking system to identify the likelihood that an ADE was associated with antibiotic use. To our knowledge, this is the first study to develop an ICD-10 code set that can be applied to administrative healthcare records to identify the incidence of antibiotic-associated ADEs. These codes can be used to investigate the incidence of recognized antibiotic-associated ADEs and to better understand the antibiotic classes posing the greatest risk of subsequent antibiotic-associated ADE-related hospital admission.

In Canada, discharge diagnoses are coded retrospectively by administrative coding specialists from patient charts and the number of discharge diagnoses coded per visit varies by province [18,19]. The way these fields are coded varies by province and country such that our results may not be generalizable outside of British Columbia. Future studies should apply the antibiotic ADE diagnostic codes identified in this work to their own administrative data to refine and validate them in other health systems. Also, while administrative data generated by acute care encounters with the healthcare system have been impactful in understanding how medications are used in the real-world setting and in generating safety signals, few confirmed ADEs are documented within them. This was demonstrated by Wickham et al. when comparing ADEs identified in administrative data with confirmed ADEs identified prospectively by care teams in three prospective cohort studies in BC (although hospital data performed the best out of the administrative datasets investigated) [12]. Thus, while our intent was to identify antibiotic-related ADEs using ICD-10 codes, these codes can only identify ADEs caused by antibiotics that were recognized and documented by the care team [20]. Ideally, they should be validated in future studies against prospectively identified and confirmed antibiotic-related ADEs. Additionally, as the diagnostic code set comprises ICD-10 codes used in hospitals, they may not capture the ADEs that patients might present with at community-based clinics (where ICD-9 codes are used). However, the code set outlined in this paper will capture the most severe and costly ADEs that patients seek care for following antibiotic use.

Future research should validate this code set by applying it to identify antibiotic-associated ADEs in records, which is currently underway in BC. This paper provides important methodological guidance on how a code set may be generated, based on real-world coded data. Our work can also be used as a starting point to develop an explicit code set for community-based ADEs, and to identify and track hospital admissions for antibiotic-related ADEs. This will allow us to identify geographic variation and time trends as new antibiotics are licensed, and prescribing guidelines vary. While, undoubtedly, the proposed code set will require iteration, it provides a starting point for improved surveillance to identify, quantify, and compare antibiotic-associated harms.

## 5. Conclusions

We used community medication dispensing data linked with subsequent hospitalization data, along with clinical review and iterative consultation, to adapt a previously published list of antibiotic-associated ADE diagnostic codes. The list shared in this work may help identify antibiotic-associated adverse drug events in hospital inpatient records in administrative data, improving the ability to capture this important outcome when assessing the benefits and risks of antibiotic administration in patient populations.

## Figures and Tables

**Table 1 antibiotics-14-00314-t001:** ADE ICD-10 causality code definitions applied to 695 unique ICD-10 codes entered for patients dispensed antibiotics in the 30 days prior to their hospital visit.

Causality Code	Definition	ICD-10 Codes (n = 695) n (%)
Likely		
A1	The ICD-10 code description includes the phrase “induced by antimicrobial medication/drug”	24 (3.5)
A2	The ICD-10 code description includes the phrase “induced by antimicrobial medication or other causes”	2 (0.3)
B1	The ICD-10 code description includes the phrase “poisoning by antimicrobial medication”	0
B2	The ICD-10 code description includes the phrase “poisoning by or harmful use of antimicrobial medication or other causes”	0
C	Adverse drug event associated with antimicrobial deemed to be very likely although the ICD-10 code description does not refer to a drug	46 (6.6)
Possible		
D	Adverse drug event associated with antimicrobial deemed to be likely although the ICD-10 code description does not refer to a drug	34 (4.9)
E	Adverse drug event associated with antimicrobial deemed to be possible although the ICD-10 code dictionary does not refer to a drug	34 (4.9)
Unlikely		
U	Adverse drug event associated with antimicrobial deemed unlikely	555 (79.9)
V	Vaccine-associated adverse event associated with antimicrobial	0

**Table 2 antibiotics-14-00314-t002:** Antibiotic-associated ICD-10 codes, categorized into “Likely”.

Consensus Score	ICD-10 Code	Description
A1	Y40.0	Penicillins
A1	Y40.1	Cefalosporins and other beta-lactam antibiotics
A1	Y40.2	Chloramphenicol group
A1	Y40.3	Macrolides
A1	Y40.4	Tetracyclines
A1	Y40.5	Aminoglycosides
A1	Y40.6	Rifamycins
A1	Y40.8	Other systemic antibiotics
A1	Y40.9	Systemic antibiotic, unspecified
A1	Y41.0	Sulfonamides
A1	Y41.1	Antimycobacterial drugs
A1	Y46.1	Oxazolidinediones
A1	T36.0	Penicillins
A1	T36.1	Cefalosporins and other beta-lactam antibiotics
A1	T36.3	Macrolides
A1	T36.4	Tetracyclines
A1	T36.5	Aminoglycosides
A1	T36.8	Other systemic antibiotics
A1	T36.9	Systemic antibiotic, unspecified
A1	T37.0	Sulfonamides
A1	T37.1	Antimycobacterial drugs
A1	T42.2	Succinimides and oxazolidinediones
A1	Z88.0	Allergy status to penicillin
A1	Z88.2	Allergy status to sulfonamides
A2	Z88.1	Allergy status to other antibiotic agents
A2	Z88.3	Allergy status to other anti-infective agents
C	D52.1	Drug-induced folate deficiency anemia
C	D61.1	Drug-induced aplastic anemia due to chemotherapy
C	E16.0	Drug-induced hypoglycemia without coma
C	G72.0	Drug-induced myopathy
C	J70.2	Acute drug-induced interstitial lung disorders
C	J70.3	Chronic drug-induced interstitial lung disorders
C	J70.4	Drug-induced interstitial lung disorders, unspecified
C	K85.3	Drug-induced pancreatitis
C	L56.0	Drug phototoxic response
C	L56.1	Drug photoallergic response
C	R50.2	Drug-induced fever
C	L27.0	Generalized skin eruption due to drugs and medicaments
C	L27.1	Localized skin eruption due to drugs and medicaments
C	L27.8	Dermatitis due to other substances taken internally
C	L27.9	Dermatitis due to unspecified substance taken internally
C	T78.2	Anaphylactic shock, unspecified
C	T78.3	Angioneurotic edema
C	T78.4	Allergy, unspecified
C	T78.9	Adverse effect, unspecified
C	T88.6	Anaphylactic shock due to adverse effect of correct drug or medicament properly administered
C	T88.7	Unspecified adverse event due to drug or medicament
C	A04.7	Enterocolitis due to Clostridium difficile
C	H91.0	Ototoxic hearing loss
C	K71.0	Toxic liver disease with cholestasis
C	K71.6	Toxic liver disease with hepatitis, not elsewhere classified
C	L51.2	Toxic epidermal necrolysis [Lyell]
C	L51.8	Other erythema multiforme
C	L51.9	Erythema multiforme, unspecified
C	D69.5	Secondary thrombocytopenia
C	D69.6	Thrombocytopenia, unspecified
C	L50.0	Allergic urticaria
C	N17.0	Acute renal failure with tubular necrosis
C	N17.1	Acute renal failure with acute cortical necrosis
C	N17.2	Acute renal failure with medullary necrosis
C	N17.8	Other acute renal failure
C	N17.9	Acute renal failure, unspecified
C	N19	Unspecified renal failure
C	D61.9	Aplastic anemia, unspecified
C	E87.5	Hyperkalemia
C	E87.6	Hypokalemia
C	I44.0	Atrioventricular block, first degree
C	I44.1	Atrioventricular block, second degree
C	R21	Rash and other nonspecific skin eruption
C	R51	Headache
C	R55	Syncope and collapse
C	R74.0	Elevation of levels of transaminase and lactic acid dehydrogenase [LDH]

**Table 3 antibiotics-14-00314-t003:** Antibiotic-associated ICD-10 codes, categorized into “Possible”.

Consensus Score	ICD-10 Code	Description
D	G62.0	Drug-induced polyneuropathy
D	L23.3	Allergic contact dermatitis due to drugs in contact with skin (TOPICAL ANTIBIOTICS)
D	L24.4	Irritant contact dermatitis due to drugs in contact with skin
D	L25.1	Unspecified contact dermatitis due to drugs in contact with skin
D	N14.1	Nephropathy induced by other drugs, medicaments, and biological substances
D	N14.2	Nephropathy induced by unspecified drug, medicament, or biological substance
D	E15	Nondiabetic hypoglycemic coma
D	K52.1	Toxic gastroenteritis and colitis
D	K71.1	Toxic liver disease with hepatic necrosis
D	K71.2	Toxic liver disease with acute hepatitis
D	K71.8	Toxic liver disease with other disorders of liver
D	K71.9	Toxic liver disease, unspecified
D	L56.2	Photocontact dermatitis [berloque dermatitis]
D	H53.1	Subjective visual disturbances
D	H53.5	Color vision deficiencies
D	H53.8	Other visual disturbances
D	H53.9	Visual disturbance, unspecified
D	L29.0	Pruritus ani
D	L29.1	Pruritus scroti
D	L29.2	Pruritus vulvae
D	L29.3	Anogenital pruritus, unspecified
D	L29.8	Other pruritus
D	L29.9	Pruritus, unspecified
D	D69.0	Allergic purpura
D	D69.2	Other nonthrombocytopenic purpura
D	I80.8	Phlebitis and thrombophlebitis of other sites
D	I80.9	Phlebitis and thrombophlebitis of unspecified site
D	K72.0	Acute and subacute hepatic failure
D	K72.9	Hepatic failure, unspecified
D	M31.0	Hypersensitivity angiitis
D	R06.0	Dyspnea
D	R17	Unspecified jaundice
D	R34	Anuria and oliguria
D	R44.1	Visual hallucinations
E	E06.4	Drug-induced thyroiditis
E	G44.4	Drug-induced headache, not elsewhere classified
E	I95.2	Hypotension due to drugs
E	M32.0	Drug-induced systemic lupus erythematosus
E	D64.2	Secondary sideroblastic anemia due to drugs and toxins
E	E03.2	Hypothyroidism due to medicaments and other exogenous substances
E	I42.7	Cardiomyopathy due to drugs and other external agents
E	N14.4	Toxic nephropathy, not elsewhere classified
E	T78.8	Other adverse effects, not elsewhere classified
E	Y57.9	Drug or medicament, unspecified
E	K71.3	Toxic liver disease with chronic persistent hepatitis
E	K71.7	Toxic liver disease with fibrosis and cirrhosis of liver
E	L51.0	Nonbullous erythema multiforme
E	L51.1	Bullous erythema multiforme
E	D69.8	Other specified hemorrhagic conditions
E	D69.9	Hemorrhagic condition, unspecified
E	E87.1	Hypo-osmolality and hyponatremia
E	E87.2	Acidosis
E	E87.3	Alkalosis
E	E87.4	Mixed disorder of acid–base balance
E	E87.7	Fluid overload
E	E87.8	Other disorders of electrolyte and fluid balance, not elsewhere classified
E	I44.2	Atrioventricular block, complete
E	I46.1	Sudden cardiac death, so described
E	I47.2	Ventricular tachycardia
E	I95.1	Orthostatic hypotension
E	I95.8	Other hypotension
E	I95.9	Hypotension, unspecified
E	K76.7	Hepatorenal syndrome
E	L26	Exfoliative dermatitis
E	R20.2	Paresthesia of skin
E	R20.3	Hyperesthesia
E	R20.8	Other disturbances of skin sensation
E	R58	Hemorrhage, not elsewhere classified

## Data Availability

The data analyzed in this study are subject to the following licenses restrictions: access to data provided by the Data Stewards is subject to approval but can be requested for research projects through the Data Stewards or their designated service providers. The following datasets were used in this study: Pharmanet and Discharge Abstracts Database. All inferences, opinions, and conclusions drawn in this publication are those of the author(s), and do not reflect the opinions or policies of the Data Steward(s).

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
