# Peer review of "ICD-10 Codes to Identify Adverse Drug Events Associated with Antibiotics in Administrative Data"

_antibiotics, 2025, doi:10.3390/antibiotics14030314_

Round 1

Reviewer 1 Report

Comments and Suggestions for Authors

Thanks for the opportunity to review this manuscript titled "ICD-10 Codes to Identify Adverse Drug Events to Antibiotics in Administrative Data." Below are my comments:

1.     The study mentions 555 “Unlikely” codes but doesn't provide specific examples. It would be helpful to include some examples to clarify which codes were considered “unlikely” to identify antibiotic-related ADEs.

2.     The study doesn’t capture ADEs from community-based clinics using ICD-9 codes. It would be helpful if the authors discussed how they plan to address this limitation.

3.     The study mentions that few ADEs are confirmed in administrative data but doesn’t explain how ADEs were confirmed.

4.     The study doesn’t compare its findings with similar research. 

Author Response

  1. The study mentions 555 “Unlikely” codes but doesn't provide specific examples. It would be helpful to include some examples to clarify which codes were considered “unlikely” to identify antibiotic-related ADEs.

Thank you for this suggestion, we have included some illustrative examples of codes classified as “unlikely” in the Results (lines 107-110).

  1. The study doesn’t capture ADEs from community-based clinics using ICD-9 codes. It would be helpful if the authors discussed how they plan to address this limitation.

This is beyond the scope of the wider study that this Technical Note contributes to. For the wider study, we will only focus on antibiotic-treated patients who required hospitalisation for antibiotic-associated ADEs. However, we thank the reviewer for this excellent suggestion for future work on community-diagnosed ADEs, which would allow for development of an analogous dataset for less severe community-managed ADEs to antibiotics. We have clarified in the Abstract (lines 20, 29), Introduction (lines 54-55), Methods (line 64) and the scope of our paper in the Discussion (line 140-144).

  1. The study mentions that few ADEs are confirmed in administrative data but doesn’t explain how ADEs were confirmed.

Thank you for highlighting this oversight, we have provided more detail of the paper being referred to in the Discussion (lines 133-137). In the referenced paper, all cases were identified by a clinical pharmacist or physicians at the point-of-care and reviewed by both. If both felt a presentation was related to an ADE, the case was considered an ADE case. All discordant cases were reviewed by an independent committee. The medical records of all diagnosed ADE cases were subsequently reviewed later to exclude any alternative diagnoses made after the initial care encounter.

  1. The study doesn’t compare its findings with similar research. 

We have outlined the previous code list (Hohl et al) that our code list was built upon and have stated in the Discussion (lines 119-121) that this is the first code list of its kind to our knowledge. We intend to apply this code set in a future study to provide initial real-world results and enable comparisons across datasets and code sets.

Reviewer 2 Report

Comments and Suggestions for Authors

The authors from British Columbia,Canada,outline methodology for developing an ICD-10 code list for antibiotic -associated ADEs. Of the 695 ICD-10 codes they identified 72 ADEs to be likely.68 were considered possible and 555 unlikely respectively.To authors knowledge this is the first study to develop an ICD- 10 code set that can be applied to administrative healthcare records to identify the incidence of antibiotic ADEs. My comments:

-in the introduction ADE is not defined. it is important for readers to distinguishes between ADE and ADR.

-I miss validation of the data

The added value of the paper is very limited. The data are  not validated and may not be generalisable outside of BC . More important is to monitor and analyze ADR than ADE to identify the incidence of antibiotic -associated ADE. From likely ADEs it is difficult to get good data regarding the incidence of ADEs.

Author Response

  1. In the introduction ADE is not defined. it is important for readers to distinguish between ADE and ADR.

We would like to thank the reviewer for highlighting this oversight, we have provided our rationale for choosing ADEs as opposed to ADRs (with definitions) to the Materials and Methods (lines 58-61).

  1. I miss validation of the data

We agree and will validate this code set in a future paper (lines 145-146). The scope of this Technical Note was to describe how we generated the code set.

  1. The added value of the paper is very limited. The data are not validated and may not be generalisable outside of BC. More important is to monitor and analyze ADR than ADE to identify the incidence of antibiotic -associated ADE. From likely ADEs it is difficult to get good data regarding the incidence of ADEs.

We thank the reviewer for their thoughts. A focus of future work will be to validate the code set that has been generated and the code set itself was reviewed by independent clinical experts and built upon a previously published code set, the validation of which was investigated (as outlined in lines 133-137). We believe that because ICD-10 codes are widely used outside of BC, this code set and the accompanying methodology for how it was derived will therefore be relevant to a wider research audience. We have included our rationale for investigating ADEs as opposed to ADRs in lines 58-61.

Reviewer 3 Report

Comments and Suggestions for Authors

This article focuses on the identification of ICD10 codes related to adverse drug events to antibiotics in administrative data. It is interesting for the health community because it tries to facilitate the correlation of adverse effects with antibiotic use. Even if the study refers to a targeted area, it can be considered a starting point for future research that can be extended worldwide.

The study is retrospective, extended on 20 years data, during 2001-2020.

It is suitable to be published in Antibiotics, but several items need to be improved to strengthen the manuscript.

The introduction is concise, and it could be correlated to more references.

The Discussion section should include more specific information about the applicability of the results in practice.

The limitations of the study should be outlined.

Did the results consider the comorbidities at the beginning of the time of hospitalization?

Please rephrase the conclusion to express more clearly the applicability of the study.

Although self citations are present, they are appropriate and they are accompanied by other relevant references.

Comments on the Quality of English Language

The English could be improved to more clearly express the research.

Author Response

  1. The introduction is concise, and it could be correlated to more references.

Thank you for the suggestion. As this is a Technical Note as opposed to a Research Manuscript, we have deliberately kept the piece brief as the scope is limited to outlining the development of the code list.

  1. The Discussion section should include more specific information about the applicability of the results in practice.

Thank for this useful suggestion. The broader intended use of this code list is stated in the Discussion (lines 121-124) however we have added more detail as to how we plan to use the code list in our study investigating antibiotic-associated ADEs in secondary care in BC (lines 145-146) and how our code set can be used for future research in this area (lines 146-153).

  1. The limitations of the study should be outlined.

The limitations of the Technical Note are outlined in the Discussion (lines 125-144).

     4. Did the results consider the comorbidities at the beginning of the time of hospitalization?

Thank you for this insightful comment. The results of this paper are the code set we generated. We will consider comorbidities into our planned cohort study that will utilise and validate this code list. The purpose of this Technical Note was to outline how clinical expertise was used to refine a previously published code list of ADEs to those that could be applicable to prior antibiotic use, using a previously published metric for outlining likelihood of causality.

    5. Please rephrase the conclusion to express more clearly the applicability of the study.

Thank you. We have added more detail in the Discussion (lines 145-153) as to how we plan to use the code list in our study investigating antibiotic-associated ADEs. We believe lines 157-160 in the Conclusion outline the broader applicability of the code list.

Round 2

Reviewer 2 Report

Comments and Suggestions for Authors

The authors responded and clarified  the reviewer comments. They improved the quality of the paper. Antibiotic associated drug events are important and should be carefully identified and investigated. The paper provides a starting point which require further validation.